# Analysis and Design of Direct Force Control for Robots in Contact with Uneven Surfaces

Antonio Rosales 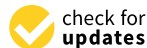 and Tapio Heikkilä *

VTT Technical Research Centre of Finland Ltd., P.O. Box 1100, FI-90571 Oulu, Finland; antonio.rosales@vtt.fi
* Correspondence: tapio.heikkila@vtt.fi

**Abstract:** Robots executing contact tasks are essential in a wide range of industrial processes such as polishing, welding, debugging, drilling, etc. Force control is indispensable in these type of tasks since it is required to keep the interaction force (between the robot and the environment/surface) within acceptable values. In this paper, we present a methodology to analyze and to design the force control system needed to regulate the force as close as possible to the desired value. The proposed methods are presented using a widely used generic contact task consisting of exerting a desired force on the normal direction to the surface while a desired velocity/position is tracked on the tangent direction to the surface. The analysis considers environments/surfaces with certain uneven characteristics, i.e., not perfectly flat. The uneven characteristic is studied using ramp or sinusoidal signals disturbing the position on the normal direction to the surface, and we present how the velocity on the tangent direction is related with the slope of the ramp or the frequency of the sinusoidal disturbance. Then, we provide a method to design the force controller that keeps the force error within desired limits and preserves stability, despite the uneven surface. Furthermore, considering the relation between the disturbance (ramp or sinusoidal) and the tangent velocity, we present a method to compute the maximum velocity for which the task can be executed. Simulations exemplifying and verifying the proposed methods are presented.

**Keywords:** robotics; force control; stability

## 1. Introduction

Robots executing contact tasks are essential to automate plenty of manufacturing processes. Regulating the force produced during the interaction between the robot and the environment is critical. There are principally two approaches to regulate the force; one is called indirect force control, since the force is regulated through motion control, i.e., changes in the position error at the end-effector, and the second one is called direct force control, since force feedback is directly compared with a desired force to calculate the robot's control input [1].

Direct force control is preferred when the application requires a precise regulation of the force. Additionally, direct force control is capable of accomplishing the contact task without damaging the environment and the robot itself [2]. However, the advantages of direct force control come at a price, since preserving stability is challenging, mainly because of the presence of unavoidable dynamics such as sensor dynamics, filters, and delays [3].

On the other hand, when direct force control techniques are implemented in industrial robots, one should design controllers that generate velocity or position inputs, since these are the standard inputs of industrial robots [4,5]. Admittance controllers are the ones having velocity/position as an output and force as an input [6]. Despite the fact that the implementation of admittance-type controllers has shown efficiency and efficacy [7], there exists a compromise between performance and stability during its design [8,9].

Recent research on force control has been focused in the design of Proportional-Integral-Derivative (PID) controllers that reach quickly the desired force with limited

overshoot. For example, the authors in [10] present a force control system based on PID that ensures asymptotic convergence of the force error to zero with small overshoot and short settling time. In [11], a force control system is presented based on PID that keeps the force within the desired value despite uncertainty in the surface's model. In [12], the authors analyze the effect of the surface's stiffness in the force control, and they present a PID controller that reaches the desired force without overshoot. Furthermore, advanced control techniques have been recently applied to regulate force. An application to medical robotics in [13] presents a force controller based on sliding mode control that ensures convergence of the force error to zero in finite time. Data-driven control is used in [14] to present a data-driven force control that ensures global convergence of the error to a steady state. Notwithstanding the prominent results presented in the mentioned references, the velocity of the robot in the tangent direction (along the surface) has not been studied, although this velocity is important since it is related to the velocity at which the task can be executed. Furthermore, a quantitative approach to design the control gains that produces a specified tolerance error is hardly discussed. The mentioned methods (estimation of the velocity of the task and a quantitative design) are relevant to practical applications required to execute the task as fast as possible and to keep the force error within acceptable limits.

In this paper, we propose a methodology to analyze and to design the force control of a robot in contact with an uneven surface. The proposed methods are presented considering the general contact task of maintaining a desired velocity along the surface (in the tangent direction to the surface) while a desired force is applied on the normal direction to the surface. This contact task properly describes applications when the priority is to regulate the force in one direction, such as polishing and assembly tasks, as well as medical applications (see [15]).

We study admittance direct force controllers with Proportional-Integral-Derivative (PID) structure to have methods suitable for industrial robots allowing velocity/position inputs and to fit our methods with the industrially accepted PID controller.

Ramp and sinusoidal signals are used to model the uneven characteristics of the surface, and the relation between the disturbances and the velocity along the surface is presented. Since we are considering only the regulation of the force in the normal direction, the magnitude of the slope (values of frequency) is bounded to avoid steep slopes producing force in a different direction than the normal one.

Then, we propose a method to compute the controller considering the performance in terms of force error and attenuation of disturbances in the normal direction. Additionally, we include the gain margin analysis to estimate how much the control gain/magnitude can be modified without creating instability. In addition, the gain margin is used to predict how much uncertainty in the stiffness the system can tolerate. Furthermore, considering the proportional relation between the velocity along the surface, and the ramp magnitude (or frequency of the sinusoidal), we provide a method to estimate the maximum velocity at which the task can be executed. The proposed methods are validated via simulations. (Preliminary results linked with this paper were presented in [16]).

The structure of the paper is the following. Section 2 presents the problem statement. The methods of analysis and design are presented in Section 3. Section 4 contains the simulations, and the conclusions are presented in Section 5.

## 2. Problem Statement

Figure 1 presents the robot in contact with the uneven surface. The robot has to execute the following task: to exert a desired force $f_d$ in the normal direction ($x$ direction) to the surface while a desired velocity $v_d$ is maintained along the surface ($y$ direction).

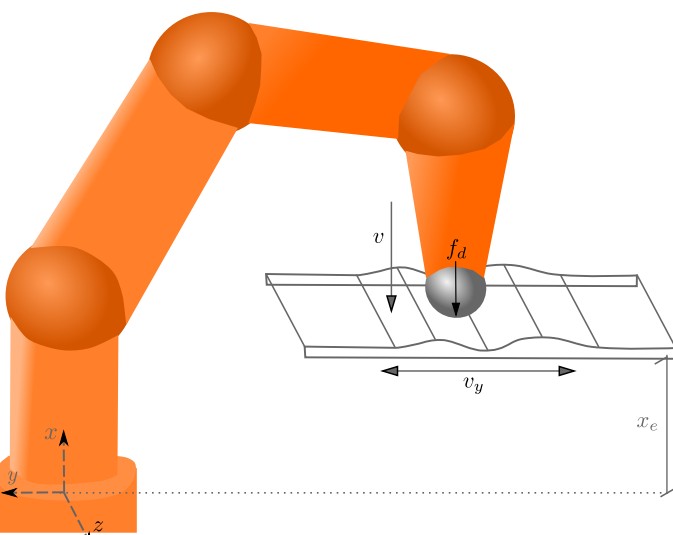

**Figure 1.** The contact task. $v$ is the velocity in the normal direction, and $v_y$ is the velocity along the surface.

During the execution of the task (see Figure 1), the following assumptions are considered. First, the end-effector of the robot is always in contact with the surface; the methods provided in this paper are not valid when the robot loses contact with the surface. Second, it is assumed that no forces are produced along the $z$-axis since the end-effector is moving along the $y$-axis. Third, the end-effector is in compliance with the surface in the $x$-direction. The compliance in the $x$-direction helps to direct most of the force produced by the curved surface to the $x$-direction; hence, the force along the $y$-axis is minimum and one can consider that the normal force is mainly defined by the force in the $x$-direction. Then, the interaction force is studied using the one-degree-of-freedom (1DOF) model presented in Figure 2, considering only the movement in the $x$ direction [6].

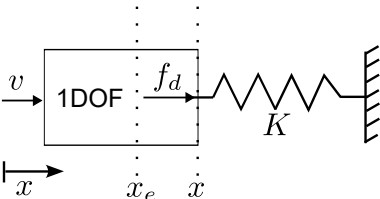

**Figure 2.** The one-degree-of-freedom interaction force model.

The force exerted by the robot on the surface is described by the following elastic model:

$$f = K(x - x_e) \tag{1}$$

where $K > 0$ is the accumulative stiffness of the tool plus the environment, $x$ is the end-effector position, and the location of the surface in $x_e$. The control objective is to design the robot's input $v$ that ensures the desired force $f_d$ is applied on the surface.

Considering the models in Figure 2 and Equation (1), the force control system presented in Figure 3 is used to study and to design the control $v$. The force control system is composed of the following blocks: $G_c(s)$ is the controller, $G_{LP}(s)$ describes the dynamics of a filter used to attenuate noise from force sensor measurements, and $G_T(s)$ corresponds to the delay produced by sensor–hardware communication. The block $K$ corresponds to the stiffness. The block named *Robot* is the single-input single-output model of the robot, and the time-constant $\tau$ defines how fast the robot's position $x$ responds to the control input $v$. The signals $f_d$, $f$, $e = f_d - f$, $v$, and $x_d$ represent the desired force, the measured force, the force error, the control signal, and a disturbance emerging in the position $x$, respectively.

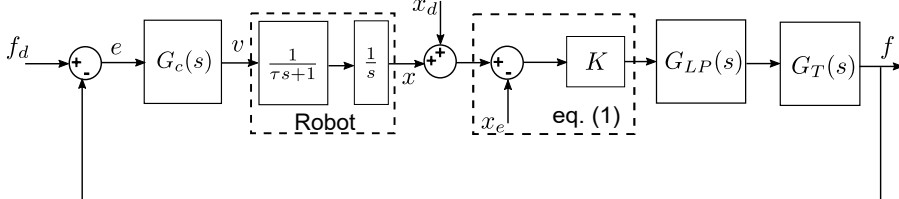

**Figure 3.** Block diagram of the force control system.

The transfer functions of the force control system in Figure 3 are as follows.

$$G_c(s) \quad = \quad K_p + K_d s + \frac{K_i}{s} \tag{2}$$

$$G_{LP}(s) \quad = \quad \frac{1}{\tau_{LP}s + 1} \tag{3}$$

$$G_T(s) \quad = \quad e^{-Ts} \tag{4}$$

where $s$ is the Laplace variable, $K_p, K_d, K_i > 0$ are control gains, $\tau_{LP}$ is the time constant of the filter, and $T$ is the time-delay value. The transfer functions and its parameters were already identified and presented in [17] by our research group. Note that the controller $G_c(s)$ is a direct force control, and it is similar to an admittance control since its input is a force and its output is a velocity.

Direct force control and admittance control have been studied and tested in industrial robots; however, during the adjustment of the gains, there exists an unavoidable compromise between performance and stability [8,9]. Furthermore, when delays and filters are included in the force control system (see Figure 3), these additional dynamics deteriorate the stability of the force control system [3]. Additionally, when disturbances emerge, the design of the gains should consider disturbance rejection as well as the stability of the system.

In this paper, we propose a method for the analysis and design of force control systems, such as the one in Figure 3, considering performance, stability margins, and robustness against disturbances. For the stability analysis, our method estimates how much the control magnitude should be modified before losing stability. Furthermore, using stability margins, we can estimate how much stiffness uncertainty the force control can handle. For the robustness analysis, we considered a disturbance ($x_d$) on the robot's position, i.e., the $x$ direction. These disturbances represent the uneven nature of the surface, and then the performance of the disturbed system is studied in terms of the force error $e$, and a method is proposed to compute the control gains that keep the error within given limits and ensure acceptable stability margins.

Ramp and sinusoidal signals are used to disturb the system. The ramp value and the frequency of the sinusoidal signal are used to define the velocity at which the task is executed. Then, from the proposed design method, the maximum velocity at which the task can be executed is estimated.

## 3. Analysis and Design of the Force Control System

In this section, we present the analysis of the force control system in Figure 3, the design method to keep the force error within acceptable limits, and how to estimate the maximum velocity at which the task can be executed.

### 3.1. Disturbance Rejection

Consider the uneven characteristics of the surface, a trapezoid with slopes of magnitude $M$ (see Figure 4a). Then, the disturbance $x_d$ is modeled using a ramp signal $X_d(s) = \frac{M}{s^2}$ with magnitude $M$, and $s$ is the Laplace variable. Note that the ramp magnitude $M$ is proportional to the velocity $v_y$ at which the task is executed. For example, when the robot

executes a linear movement from point A to point B, the bigger the velocity $v_y$, the bigger the magnitude $M$ of the ramp in the $x$ direction.

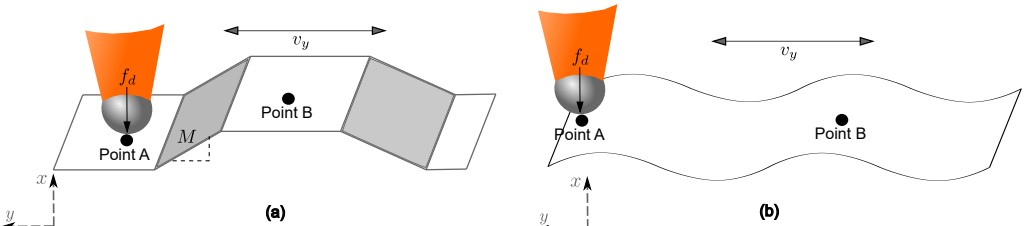

**Figure 4.** The uneven characteristics of the surface. (**a**) Ramp shape and (**b**) sinusoidal shape.

The performance of the force control system in Figure 3 is analyzed using the steady-state error $e_{ss} = \lim_{s \to 0} sE(s)$, where $E(s)$ is the force error $e$ in the Laplace domain; then

$$e_{ss} = \lim_{s \to 0} s \frac{KG(s)}{1 + G_c(s)KG(s)G_{LP}(s)G_T(s)} X_d(s) \qquad (5)$$

where $G_c = K_p + K_d s + \frac{K_i}{s}$ is the controller, $G(s) = \frac{1}{s(\tau s + 1)}$ is the transfer function of the robot, $G_{LP}(s)$ is the filter, $G_T$ is the transport delay, $X_D(s)$ is the external disturbance, and $K$ is the stiffness.

Considering the ramp disturbance $X_d(s) = \frac{M}{s^2}$, the control $G_c(s) = K_p + K_d s + \frac{K_i}{s}$ ensures zero steady-state error $e_{ss}$ [18]. However, the integral term $\frac{K_i}{s}$ has a drawback since it produces a sluggish and oscillatory response.

On the other hand, when the controller is $G_c(s) = K_p + K_d s$, the steady-state error is $e_{ss} = \frac{M}{K_p}$, but it can be reduced by incrementing the control gain $K_p$ (or the control magnitude $|G_c(s)|$). In Figure 5, the curves for different values of $e_{ss}$ are presented. These curves are obtained from $e_{ss} = \frac{M}{K_p}$ using different ramp values and gains $K_p$.

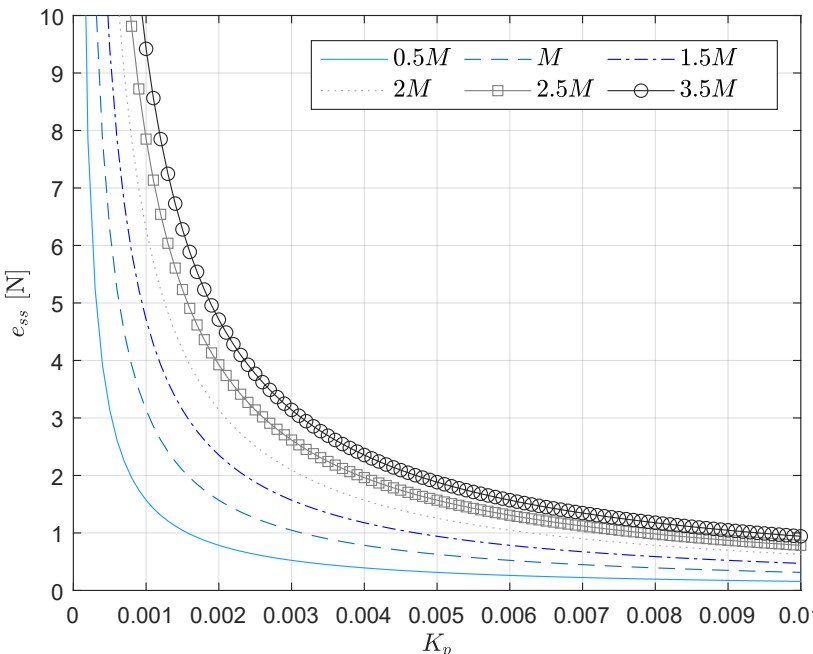

**Figure 5.** Curves of steady-state error $e_{ss}$ in terms of control gain $K_p$ and ramp value $M$.

The curves in Figure 5 represent a design tool considering the disturbance ramp magnitude $M$ (proportional to the velocity $v_y$ at which the task is executed) and force error

via steady-state error. If the magnitude $M$ is known/estimated, one can obtain the control gain $K_p$ that produces the steady-state error $e_{ss}$ presented in the curves, and vice versa, if a certain steady-state error $e_{ss}$ is desired, one can choose the gain $K_p$ producing this error.

Note than the stiffness value $K$ does not appear in disturbance analysis presented in this section. However, the value of $K$ matters when it is big since the stability of the force control system may be compromised. Furthermore, one should be careful when the gain $K_p$ is selected, since a big $K_p$ value may affect the stability too.

Time-Varying Disturbances

Another way to study the uneven characteristics of the surface is using sinusoidal disturbances $x_d(t) = \sin(\omega t)$ (see Figure 4b). In this case, the velocity $v_y$ at which the task is executed is proportional to the frequency $\omega$ of the disturbance in the $x$ direction. For example, the faster the end-effector is moving on the surface from point A to point B, the higher the frequency $\omega$ of the sinusoidal disturbance. The proportionality relation depends on the wavelength $\lambda$ of the sinusoidal, i.e., $v_y = \lambda(\omega/2\pi)$.

When a sinusoidal disturbance appears, the steady-state error $e_{ss}$ cannot be used to analyze the force error, and the analysis presented in Section 3.1 is not valid. However, one can find a relation between the magnitude of the time-varying disturbance and the control gain/magnitude.

Consider the following transfer function,

$$\frac{F(s)}{X_d(s)} = \frac{KG_{LP}(s)G_T(s)}{1 + G_c(s)G(s)KG_{LP}(s)G_T(s)},$$

from the disturbance $x_d$ to $f$ ($f_d = 0$). From [18], this transfer function can be approximated by $\frac{F(s)}{X_d(s)} \approx \frac{1}{G_c(s)G(s)}$. Then, the disturbance $X_d(s)$ can be attenuated by increasing the magnitude of $G_c(s)$, since the goal is to have a magnitude,

$$|F(s)|_{s=j\omega} = \frac{|X_d(s)|_{s=j\omega}}{|G_c(s)|_{s=j\omega}|G(s)|_{s=j\omega}}, \tag{6}$$

as close as possible to zero. Again, one should be aware of stability when the magnitude of the controller increases.

Note that the computation of the magnitude $|F(j\omega)|$ depends on the parameters of the controller $G_s$, the dynamics of the robot $G$, and the frequency $\omega$. Whenever the mentioned parameters are available, one can obtain curves similar to those presented in Figure 5; an example of the curves is presented in Section 4.

### 3.2. Stability Analysis

In this section, the relative stability analysis of the force control system in Figure 3 is performed. Then, we find how much the control gain/magnitude $K_p$ (also stiffness $K$) can be incremented without damaging stability.

### 3.2.1. Stability in Terms of $K$

The gain margin is computed using the open-loop transfer function $L(s)$. For the force control system in Figure 3, the transfer function $L(s)$ is as follows (see [18]):

$$L(s) = KG_c(s)G(s)G_f(s)G_d(s). \tag{7}$$

Considering $s = j\omega$, the gain margin is obtained using the magnitude of $L(j\omega)$,

$$|L(j\omega)| = K|G_c(j\omega)||G(j\omega)||G_f(j\omega)||G_d(j\omega)|,$$

where $\omega$ is the frequency associated with the frequency response of $L(s)$. Note that the magnitude $|L(j\omega)|$ is directly proportional to the gain $K$ independent of the frequency $\omega$.

From the definition of the gain margin [18], the gain margin is the biggest increment of magnitude $|L(j\omega)|$ that conserves stability. The condition for stability is $|L(j\omega)| < 1$, and this condition can be tested in the following way. First, a multiplicative gain $K_{GM}$ is added to $|L(j\omega)|$, and second, $K_{GM}$ is increased until the stability condition is violated, i.e., $K_{GM}|L(j\omega)| \geq 1$ [19]. The magnitude $|L(j\omega)|_{test}$,

$$|L(j\omega)|_{test} = K_{GM}K|G_c(j\omega)||G(j\omega)||G_f(j\omega)||G_d(j\omega)|,$$

is used to check the gain margin, and the stability is ensured if $|L(j\omega)|_{test} < 1$. Note that the stability condition will be violated for certain $K_{GM} = K_{GM_{max}}$ producing $|L(j\omega)|_{test} = 1$, and the value of the gain margin will be $K_{GM_{max}}$.

One can observe that the term $K_{GM}K$ affects the whole magnitude $|L(j\omega)|_{test}$. Therefore, when the gain margin $K_{GM_{max}}$ is known, one can use $K_{GM_{max}}$ to estimate the maximum increment/change in stiffness $K$ that maintains the stability.

### 3.2.2. Stability in Terms of $K_p$

For the computation of the gain margin in terms of $K_p$, the stiffness $K$ is considered constant and the frequency response $G_c(j\omega)$ is divided in real and imaginary parts, $G_c(j\omega) = K_p + \left(\frac{K_d\omega^2 - K_i}{\omega}\right)j$. Adding the multiplicative gain $K_{GM}$ to $G_c(j\omega)$, the magnitude $|G_c(j\omega)|_{K_{GM}}$ is defined as

$$
\begin{aligned}
|G_c(j\omega)|_{K_{GM}} &= K_{GM}|G_c(j\omega)| = \left| K_{GM}K_p + \left( \frac{K_{GM}K_d\omega^2 - K_{GM}K_i}{\omega} \right)j \right|, \\
&= K_{GM}\left| K_p + \left( \frac{K_d\omega^2 - K_i}{\omega} \right)j \right|.
\end{aligned}
$$

Note that the gain $K_{GM}$ is directly proportional to the control magnitude $|G_c(j\omega)|_{K_{GM}}$ or directly proportional to each control gain $K_p, K_d$, and $K_i$.

Using $|G_c(j\omega)|_{K_{GM}}$, the transfer function to test and compute the gain margin is

$$|L(j\omega)|_{test} = K_{GM}|G_c(j\omega)|K|G(j\omega)||G_f(j\omega)||G_d(j\omega)|,$$

where $K_{GM}$ represents an increment/change in the magnitude $|G_c(j\omega)|$. Since the stability condition is $|L(j\omega)|_{test} < 1$, the gain margin is the gain $K_{GM} = K_{GM_{max}}$ that produces $|L(j\omega)|_{test} = 1$. Therefore, the gain margin computation gives an estimate of how much one can modify the control gain/magnitude without producing instability.

### 3.3. Design Method

Considering the stability analysis presented in the preceding section and the design curve in Figure 5, one can observe a compromise between stability and error attenuation. Selecting a big value of control magnitude $|G_c(j\omega)|$ might result in an acceptable force error but this magnitude may deteriorate the stability.

The proposed design method for the controller $G_c = K_p + K_d s$ is obtained, when one includes in the curves of Figure 5 the maximum value of magnitude/gain $K_{GMmax}$ that preserves stability (see Figure 6). The set of gains $K_p$ presented in Figure 6 is selective since it considers only the gains $K_p$ that guarantee stability. The values of $e_{ss}$ corresponding to the gain $K_{GMmax}K_p$ are the minimum values one can have in the force error considering the disturbance magnitude $M$ and the stability margin. Therefore, the curves in Figure 6 provide a better design method for the controller $G_c(s)$ compared with the curves in Figure 5.

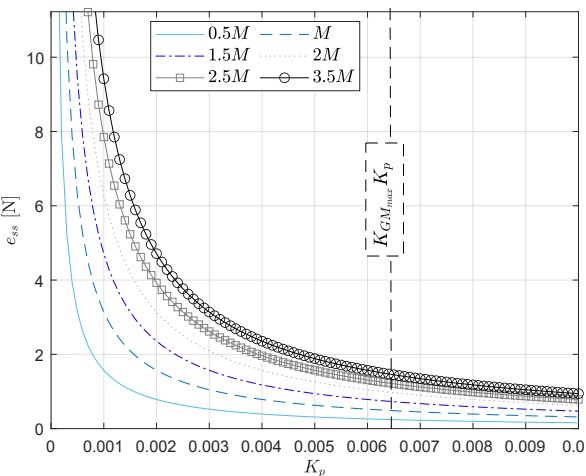

**Figure 6.** Design method: steady-state error $e_{ss}$ in terms of control gain $K_p$ and ramp value $M\,K_p$ considering gain margin.

Selecting Gains of $G_c$ Considering Stiffness $K$

From Equation (7), one can observe how the product of $K$ and $|G_c(j\omega)|$ affects the gain margin. Therefore, when the control gains in Equation (2) are selected/adjusted, one should consider the value of stiffness $K$ in order to preserve the relative stability of the system.

Assume the stiffness $K$ can be modified/adjusted (adding elasticity to the end-effector using a spring). Then, from Equation (7), the magnitude $|L(j\omega)|$ contains two adjustable terms (its parameters are at hand); the first one is the controller $G_c$ and the second one is the stiffness $K$.

If the goal is to preserve a desired gain margin, one should keep the magnitude $|L(j\omega)|$ as close as possible to its value associated with the desired gain margin. Therefore, when the control gains are adjusted (or the stiffness is adjusted), one should keep a balance between the magnitude of the controller $G_c$ and the value of stiffness $K$. For example, if the magnitude of $G_c$ increases, one should balance/compensate this change with a decrease in $K$ to preserve the desired magnitude $|L(j\omega)|$ associated with the desired gain margin.

Therefore, in order to preserve a stable contact force, the following relations between the stiffness and the controller exist:

- For a rigid surface/environment, a compliant controller is needed, i.e., $K >> |G_c|$.
- For a compliant surface/environment, a rigid controller is needed, i.e., $K << |G_c|$.

### 3.4. Estimation of the Maximum Velocity at Which the Task Is Executed

Assume $e_{max}$ is the maximum tolerable force error in the force control system in Figure 3. Then, all the errors $e_{ss} < e_{max}$ are acceptable.

On the other hand, when one selects the maximum gain $K_{GMmax}K_p$ from Figure 6, this gain is the critical gain since it corresponds to the case $|L(j\omega)|_{test} = 1$. In practice, one should avoid having a critical gain, since the system may have an oscillatory response, and a small disturbance may cause instability. Therefore, the selection of the controller gain should be $K_p < K_{GMmax}K_p$.

Considering the maximum error $e_{max}$ and the recommended selection of the gain $K_p < K_{GMmax}K_p$, it is possible to obtain a more selective set of gains $K_p$ from Figure 6. In Figure 7, certain values for $e_{max}$ and $K_p < K_{GMmax}K_p$ are presented. One can observe a selective set of gains $K_p$ defined by the limits $e_{max}$ and $K_p < K_{GMmax}K_p$. The mentioned region contains the set of gains that ensure an acceptable force error and a safer response, since $K_p$ is far from the critical gain. Additionally, there are curves corresponding to different values of $M$, and a maximum value $M_{max}$ can be obtained from the region. In Figure 7, $M_{max} = 1.5M$. Since $M$ is proportional to the velocity at which the task is executed (see Section 3.1), the maximum velocity can be estimated from $M_{max}$.

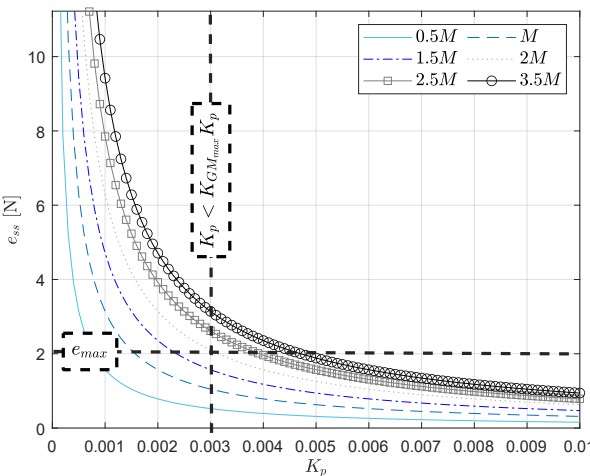

**Figure 7.** Curves including maximum tolerable error $e_{max}$ and recommended $K_p < K_{GMmax}K_p$.

## 4. Simulations

Firstly, using the *pidTuner* of Matlab, we tuned three controllers $G_c(s)$, a Proportional (P), Proportional-Derivative (PD), and Proportional-Integral-Derivative (PID) to have the same settling time $t_s \approx 0.8$ seconds and overshoot of 15%. The gains of the mentioned controllers are presented in Table 1.

**Table 1.** Control gains.

|     | $K_p$ | $K_d$ | $K_i$ | $t_s$ | Overshoot |
| --- | --- | --- | --- | --- | --- |
| P | 0.002229 | 0 | 0 | 0.88 s | 14.8% (7.4 N) |
| PD | 0.004211 | 0.001111 | 0 | 0.71 s | 14.4% (7.2 N) |
| PID | 0.0053947 | 0.0005154 | 0.01411 | 0.88 s | 15.4% (7.7 N) |

For the simulation, the force control system in Figure 3 is built in Simulink, and the simulation is executed using the solver ode1(euler) with a fixed sampling time of 1 millisecond. The parameters of the force control system used in the simulation are $\tau = 0.05$, $T = 0.008$, $K = 3000$, and $\tau_{LP} = 0.05$. We test a step input of 50 N at 30 s. Figure 8 shows the time response; one can see that the settling time and overshoot are similar but the PD controller is faster than P and PID.

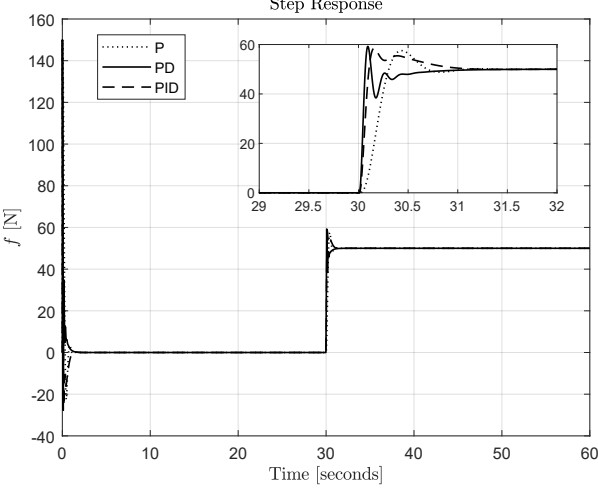

**Figure 8.** Simulation results: force $f$ with P, PD, and PID.

*4.1. Disturbance Rejection*

Consider the controller $G_c = K_p + K_d s$ with the parameters of Table 1, and a ramp disturbance of magnitude $M = 3.14 \times 10^{-3}$ emerging at $t = 20$ s. Then, the simulation is performed, and Figure 9 presents the disturbance $x_d(t)$ and the force error $e$ from the simulation. Note that the disturbance $x_d(t)$ produces a force error of $e_{ss} \approx 0.74$ [N]. This error corresponds with that estimated theoretically using $M = 0.00314$ and $K_p = 0.004211$, i.e., $e_{ss} \approx \frac{0.00314}{0.004211} \approx 0.74$.

Considering the design curves of Figure 6, if we want to reduce the error $e_s s$, we need to increment the gain $K_p$. Then, increasing the gain $K_p$ to $K_{p_a} = K_p + 0.002$ and $K_{p_b} = K_p + 0.004$, the simulation of the force control system using these gains is performed, and the resulting error is shown in Figure 9. One can observe that $e_{ss}$ is reduced when the value of $K_p$ is increased.

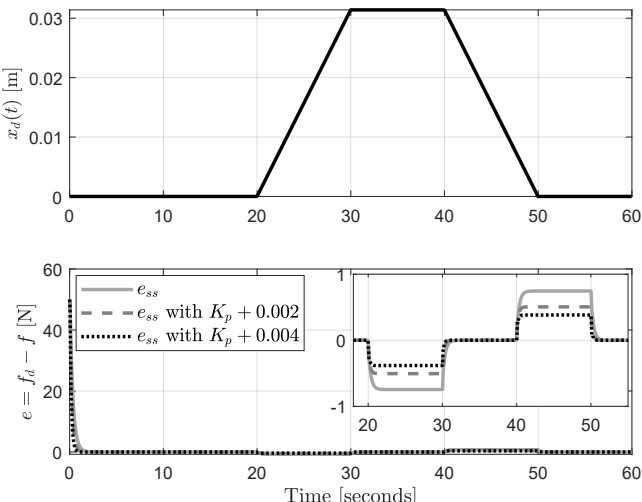

**Figure 9.** Force error $e_{ss}$ in presence of ramp disturbance and disturbance $x_d(t)$.

Time-Varying Disturbances

Considering a sinusoidal disturbance $x_d(t) = 0.01 \sin(0.628t)$ emerging at $t = 20$ s, and the controller $G_c = K_p + K_d s + \frac{K_i}{s}$ with the gains in Table 1, the simulation of the force control system is performed. The resulting force error $e$ is presented in Figure 10 for different values of control magnitude. One can observe that the error decreases when the gains of magnitude of $G_c(s)$ increase, as expected from Equation (6).

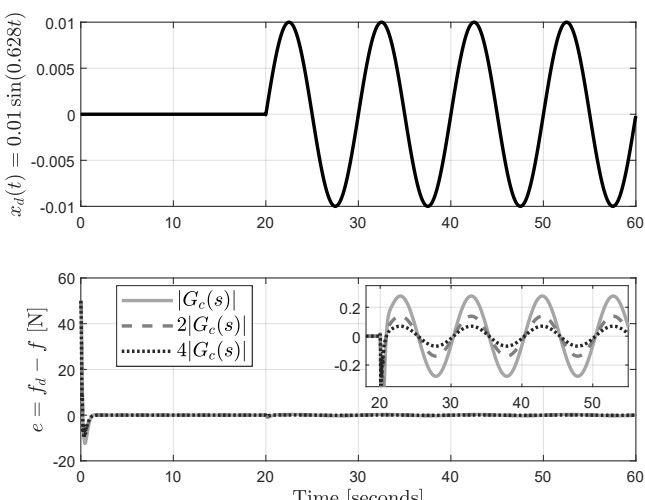

**Figure 10.** Force error $e_{ss}$ in presence of sinusoidal disturbance $x_d(t) = 0.01 \sin(0.628t)$ and disturbance $x_d(t)$.

### 4.2. Stability

In this subsection, we compute the gain margin of the force control system in Figure 3. Figure 11 presents the Bode plot of the system with controller $G_c = K_p + K_d s$ and $G_c = K_p + K_d s + \frac{K_i}{s}$. The gain margin is computed in the crossing of the magnitude Bode plot with zero decibels (see [18]); this intersection is indicated with an arrow in Figure 11. For the PD controller, the gain margin is equal to 11.1 dB, which is equivalent to 3.6 ($10^{(11.1/20)}$) in magnitude. For the PID controller, the gain margin is equal to 16 dB, which is equivalent to 6.3 ($10^{(16/20)}$) in magnitude. This gain margin represents the maximum value of the control gain (magnitude) that one can use without compromising the stability of the system. From the analysis presented in Section 3.3, the gain for the system with the PD controller is $K_{GM_{max}} = 3.6$, and for PID, the magnitude is $K_{GM_{max}} = 6.3$. From the design curves in Figure 5, the control gain $K_p$ can be increased until 3.6 times without losing stability when the PD controller is used.

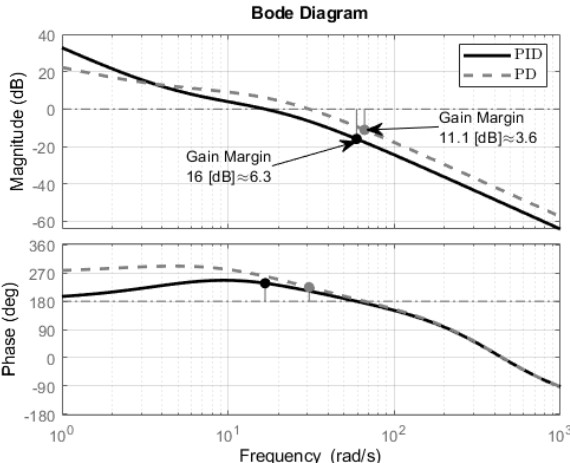

**Figure 11.** Bode plot and stability margins.

Now, the simulation is executed with $G_c = K_p + K_d s$, and two values of $K_p$, i.e., $K_{p_1} = 3K_p$ and $K_{p_2} = 3.45K_p$. The resulting force error $e = f_d - f$ is presented in Figure 12a,b. One can observe that the higher the gain, the more oscillations in the force error $e$. Furthermore, a gain of $K_{p_3} = 3.6K_p$ was tested, but these results are not presented in Figure 12, since this gain produces instability. Note that the simulation results correspond with the gain margin presented in Figure 11, since oscillations appear when the control gains are closer to the gain margin.

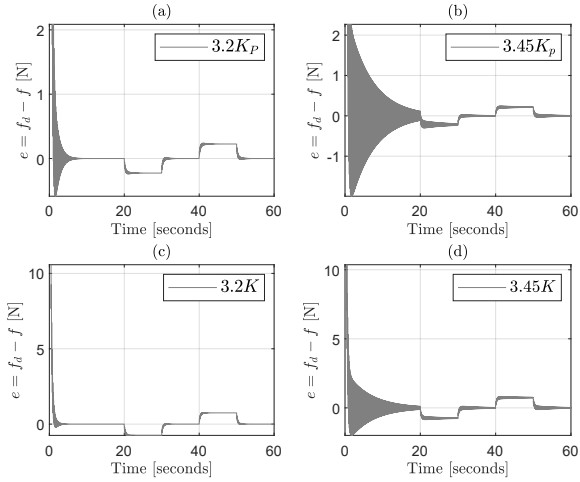

**Figure 12.** Force error $e = f_d - f$ with PD controller. (**a,b**) The control gain $K_p$ increases. (**c,d**) The stiffness $K$ increases.

In accordance with Section 3.3, one can relate the gain margin shown in Figure 11 with the maximum value of stiffness that preserves stability. This value is $K_{max} \approx K_{GM_{max}} K \approx 3.6\,K$. Figure 12c,d present the simulation of the force control system with two different values of stiffness, $K_a = 3.2\,K$ and $K_b = 3.45\,K$, where $K = 3000$. The resulting force error $e$ presents oscillations when $K$ increases. For stiffness values higher than $3.5\,K$, the system lost stability. This unstable case is not presented in Figure 12 for visibility purposes. The simulation matches with the estimated gain margin, since oscillations/instability appear when the value of stiffness $K$ approaches/reaches the gain margin.

### 4.3. Design Method

In Section 4.2, a gain margin of 3.6 was obtained. Considering a constant stiffness $K$, the maximum gain $K_{p_{max}}$ that preserves stability is $K_{p_{max}} \approx 3.6 K_p$. Then, adding this maximum gain $K_{p_{max}} \approx 3.6 K_p$ into the design curves in Figure 6, one can obtain the gain that keeps the error within desired values while preserving stability. Figure 13 shows the design curves, including the stability margin bound $K_{p_{max}} \approx 3.6(0.004211) \approx 0.0152$. The curves are obtained considering a value of $M = 0.00314$.

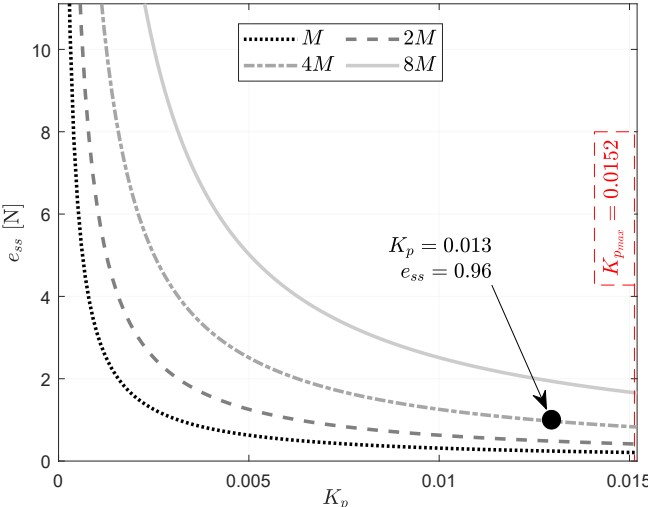

**Figure 13.** Design method: steady-state error $e_{ss}$ in terms of control gain $K_p$ and ramp value $M$, considering gain margin.

The curves in Figure 13 can be used to find the control gain $K_p$ that gives a desired force error. Assume the magnitude of the disturbance is provided, for example, $4M$. Then, if one selects a control gain of $K_p = 0.013$, the expected steady-state error is $e_{ss} \approx 0.96$ N (see the curve $4M$ in Figure 13).

Considering the disturbance of magnitude $4M$ and the controller $G_c = K_p + K_d s$ with $K_p = 0.013$, the force control system is simulated. Figure 14 presents the disturbance and the resulting force error $e$. Note that the force error $e$ in Figure 14, when $K_p = 0.013$, corresponds to the value of $e \approx 1$ in the curves of Figure 13.

Since $K_p = 0.013$ is close to $K_{p_{max}}$, a small increase in $K_p$ can cause oscillations and instability in the system. The presented technique can be combined with root-locus analysis to make an adjustment of $K_p$ to have a desired damping. Figure 15 presents the root-locus of the force control system, computed from the open-loop transfer function $L(s)$ in Equation (7) with $G_c = K_p + K_d s$ and using a second-order Padé approximation of $G_T(s)$, i.e., $G_T(s) = \approx \frac{N_r(sL)}{D_r(sL)}$, where $N_r(sL) = \sum_{k=0}^{r} \frac{(2r-K)!}{k!(r-k)!}(-sT)^k$, $D_r(sL) = \sum_{k=0}^{r} \frac{(2r-K)!}{k!(r-k)!}(sT)^k$, $T$ is the value of the delay, and $r$ is the order of the approximation (see [18]). Figure 15 contains three markers showing the critical gain value 6.3 (similar to the gain margin), the gain 3.08 corresponding to $K_p = 0.013$, and the gain value 0.231 corresponding to a damping factor of 0.7.

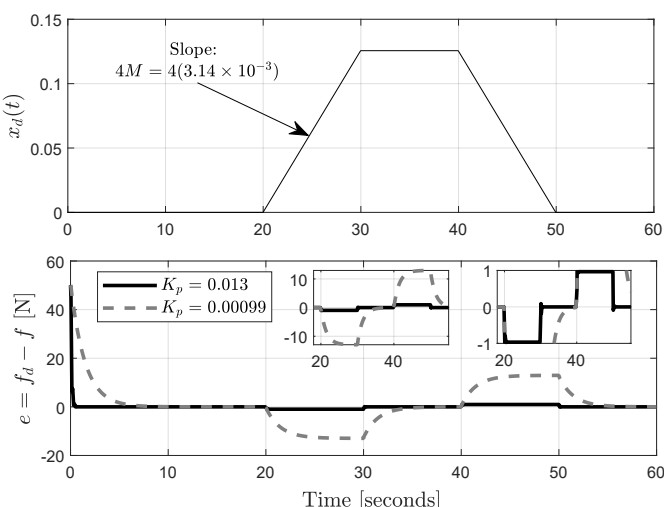

**Figure 14.** Force error $e_{ss}$ in presence of ramp disturbance and disturbance $x_d(t)$.

Figure 14 presents the simulation when $K_p$ is adjusted by the gain value 0.231, i.e., $K_p = 0.231 \times 0.004422 = 9.9 \times 10^{-4}$. One can see that the force error $e$ has a damped response when $K_p = 9.9 \times 10^{-4}$ but the value of $e$ is bigger than the case when $K_P$ is close to the gain margin value. Therefore, one can observe the compromise between force regulation and stability.

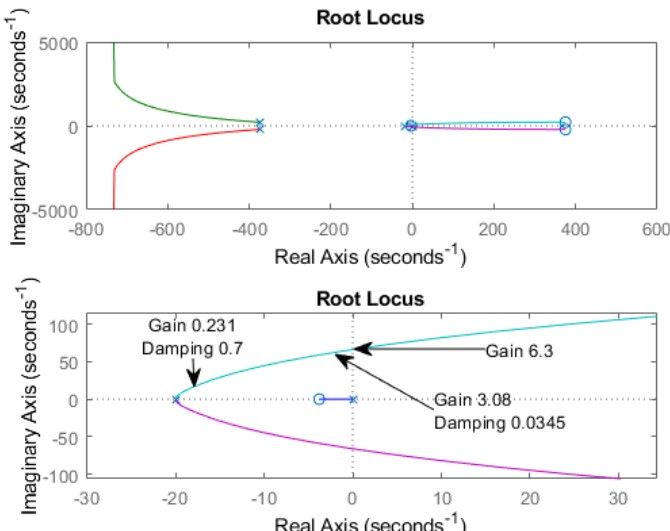

**Figure 15.** Root-locus analysis computed with $L(s)$ in Equation (7). Bottom part: zoom-in around the origin.

Time-Varying Disturbances

The design curves, such as those in Figure 5, for the case of time-varying disturbance are computed as follows. Considering the disturbance $x_d(t) = 0.01\sin(0.628t)$ and the controller $G_c = K_p + K_d s + \frac{K_i}{s}$ with the gains presented in Table 1, Equation (6) can be used to obtained the curves presented in Figure 16. These curves are computed for a fixed frequency of 0.628 and different sinusoidal amplitudes (from 0.005 to 0.08). The horizontal axis represents the controller magnitude, and the vertical axis represents the deviation of the force from its desired value.

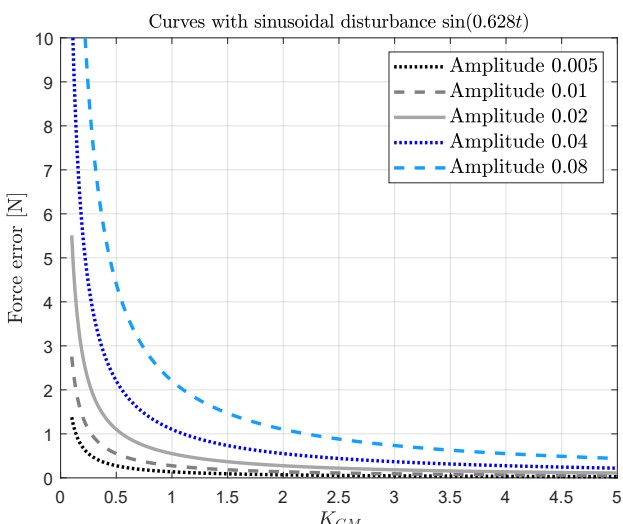

**Figure 16.** Curves of steady-state error $e_{ss}$ in terms of control gain $K_p$ and sinusoidal amplitude.

The simulation of the force control system with disturbance $x_d(t) = \sin(0.628t)$ for different amplitudes (0.01, 0.02, 0.04, and 0.08) is performed, and the results are presented in Figure 17. Note that the deviation of the force $f$ with respect to the reference $f_d = 50$ N corresponds with the deviation predicted by the curves in Figure 16 for the case $K_{GM} = 1$, since the control magnitude was not changed. Specifically, Figure 16 predicts an approximated deviation of 0.25, 0.5, 1, and 2, when a sinusoidal disturbance with magnitude 0.01, 0.02, 0.04, and 0.08, respectively, appears in the system. This prediction matched with the simulation results presented in Figure 17; see the zoom-in of the figure.

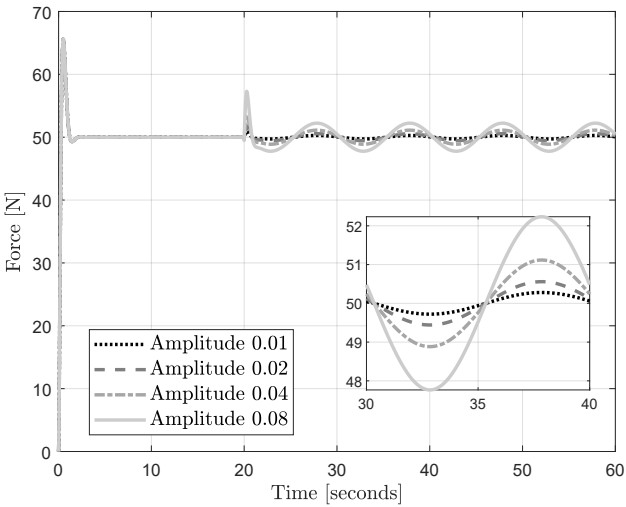

**Figure 17.** Simulation of the force control system: sinusoidal disturbance with variable amplitude.

When the amplitude of the sinusoidal disturbance is fixed, one can obtain curves similar to those presented in Figure 16 for several frequency values. Figure 18 presents the mentioned curves, when the sinusoidal disturbance has an amplitude of 0.01 and different frequency values. Then, whenever the amplitude and frequency are known, the curves in Figure 18 can be used to tune the control magnitude in accordance with the desired force error.

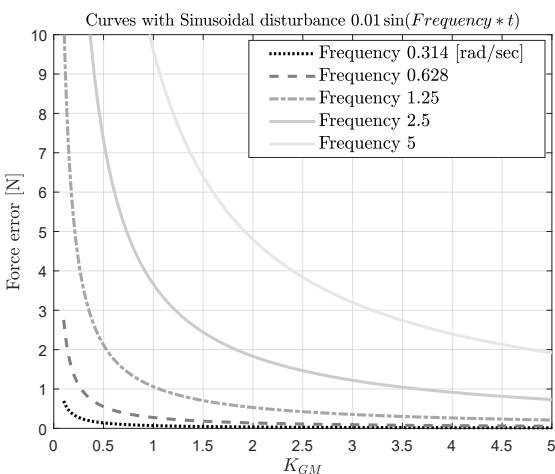

**Figure 18.** Curves of steady-state error $e_{ss}$ in terms of control magnitude $K_{GM}$ and sinusoidal frequency.

Figure 19 presents the results of the simulation of the force control system with the sinusoidal disturbance of amplitude of 0.01 and different values of frequency. One can see that the simulation results matched with the predicted force error of the design curves in Figure 18. For a control magnitude of one, i.e., $K_{GM} = 1$, the predicted force errors are 0.5, 1, 3.5, and 10 [N], when a disturbance with frequency 0.628, 1.25, 2.5, and 5 rad/s, respectively, emerges. These force errors are similar to the ones obtained in the simulation presented in Figure 19; see the zoom-in. Note that the gain margin of 6.3 (see Figure 11) must be considered during the selection of the control magnitude $K_{GM}$ in the curves presented in Figures 16 and 18.

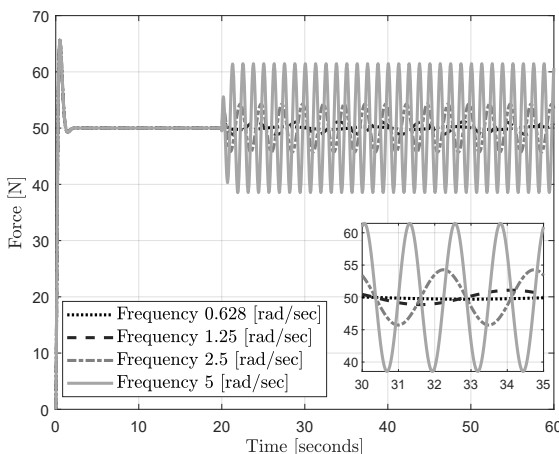

**Figure 19.** Simulation of the force control system: sinusoidal disturbance with variable frequency.

### 4.4. Maximum Speed along the Surface

The maximum speed at which the task can be executed is computed as follows. Consider that the maximum force error tolerated by the system is $e_{max} = 3$ N, and controller gain $K_p = 0.005$ (close to the one in Table 1). Then, locating these values in the design curves of Figure 13, one obtains a region containing the possible values of disturbance magnitude $M$ that can be compensated (see Figure 20). The maximum value of $M$ is the one located in the upper right corner of the rectangular region delimited by $e_{max} = 3$ N and $K_p = 0.005$. Note that the maximum value is $4M$. Then, the maximum speed at which the task can be executed is computed as $4M = 4 \times 0.00314 = 0.0126$ m/s.

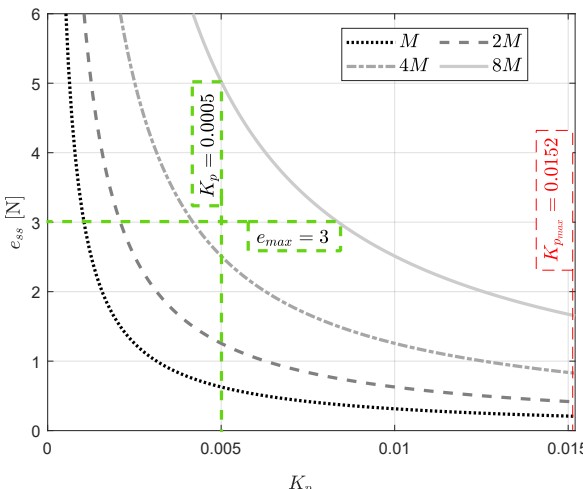

**Figure 20.** Computing the maximum speed using the curves of steady-state error $e_{ss}$ in terms of control gain $K_p$ and ramp value $M$.

For the case of time-varying disturbances, the maximum speed at which the task can be executed is computed using the curves presented in Figure 18. Considering a maximum error of $e_{max} = 3$ N and a selection of control magnitude $K_{GM} = 1.5$, the maximum velocity can be estimated with the frequency value associated with the closest curve to the upper right corner of the rectangular region delimited by $e_{max} = 3$ N and $K_{GM} = 1.5$. In Figure 21, this frequency value is 2.5 rad/s. Then, assuming a wavelength $\lambda = 0.1$ m and considering the maximum frequency of 2.5 rad/s, the maximum velocity is $0.1 \times (2.5/2\pi) = 0.0199$ m/s.

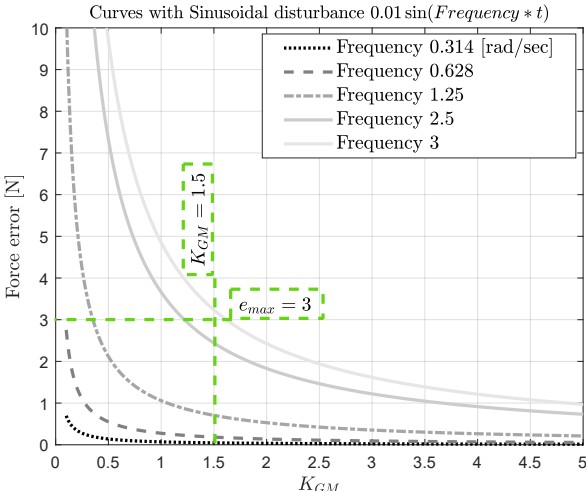

**Figure 21.** Computing the maximum speed using the curves of steady-state error $e_{ss}$ in terms of control magnitude $K_{GM}$ and disturbance frequency.

## 5. Conclusions

The methods of analysis and design presented in this paper are useful to keep the force error within desired limits while guaranteeing stability. Furthermore, the presented design curves can be used to estimate the maximum velocity at which the task can be executed. Since this method is model-based, its application requires certain knowledge about the disturbances acting on the system, such as maximum magnitude and frequency. However, these parameters might be available in practical applications or not difficult to estimate. The simulations presented in the paper verify the effectiveness of the proposed methods.

**Author Contributions:** A.R. and T.H. contributed equally to this paper. All authors have read and agreed to the published version of the manuscript.

**Funding:** This research was funded by Business Finland and VTT Technical Research Centre of Finland Ltd.

**Institutional Review Board Statement:** Not applicable.

**Informed Consent Statement:** Not applicable.

**Data Availability Statement:** No new data were created or analyzed in this study. Data sharing is not applicable to this article.

**Conflicts of Interest:** The authors declare no conflict of interest.

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
