# Peer review of "Analysis and Design of Direct Force Control for Robots in Contact with Uneven Surfaces"

_applsci, doi:10.3390/app13127233_

Round 1

Reviewer 1 Report

It is my opinion that the paper is very well written and I really enjoyed reading it. My only question/suggestion would be about the adjusted gain value presented on page 12, lines 307-309. I guess the decimal point in 0.231 ∗ 004422 was lost by mistake.

Reviewer 2 Report

In this paper, the authors proposed a methodology to analyze and to design the force control of a robot in contact with an uneven surface.

If there are unknown surfaces that are not parallel to the horizontal, the forces are not one-dimensional. A partial force will also act for other direction. 

How did you consider this physical property? This point must be clearly explained in the paper.

If you only discuss the block digarm shown in Fig.3 for one-dimensional, this seems to me just a practic of control engineering and not a research work.

Why have you used the controller shown in Fig.3?

Why you have not consider the elastic force in Eq.(2)?

The motivation and originality of the work must be clearly written in the paper.

Reviewer 3 Report

The paper is interesting and devoted to solving a practical problem. It presents an original method of tuning the robot arm control system. For a better understanding, it is advisable to make some minor adjustments.

 The article contains the following statements:

 - “The control objective is to design the robot’s input “v”but this speed is not shown in Fig. 1 - a correction of the drawing is recommended.  - “The velocity of the robot on x direction is defined by the first-order system” –You should justify why – this is a significant simplification.

 “Fig. 12 (c)-(d) presents the simulation of the force control system with two different 276 values of stiffness K, 3.2K and 3.45K. The resulted force error e presents oscillations when 277 K increases. For values of K higher than 3.5K, the system lost stability – the mathematical notation should be corrected – the current notation means that K=3.2K and that K>3.5K – these are not correct notations, the markings should be modified.

Round 2

Reviewer 2 Report

This paper does not seem to answer my question.

Why do you only consider forces in one direction?

If the arm is moved along a curved surface, the force is not one-dimensional.

Figure 1 is misleading. If the robot has only one degree of freedom, this may not be a problem. But why would it have a first-order delay system?

What kind of actuators are used in the robot? Usually, when electric motors are used, it is a second-order delay system.

Round 3

Reviewer 2 Report

The revised paper answered my question.